# Assessing Patients with Alpha-1 Antitrypsin Deficiency for Corneal Refractive Surgery: A Review and Clinical Experience

**DOI:** 10.3390/jcm11144175

**Published:** 2022-07-19

**Authors:** Majid Moshirfar, Neil Kelkar, Yasmyne C. Ronquillo, Phillip C. Hoopes

**Affiliations:** 1Hoopes Vision Research Center, Hoopes Vision, 11820 S. State St. #200, Draper, UT 84020, USA; yronquillo@hoopesvision.com (Y.C.R.); pch@hoopesvision.com (P.C.H.); 2John A. Moran Eye Center, University of Utah School of Medicine, Salt Lake City, UT 84132, USA; 3Utah Lions Eye Bank, Murray, UT 84107, USA; 4College of Medicine, University of Arizona College of Medicine-Phoenix, Phoenix, AZ 85004, USA; nkelkar@email.arizona.edu; 5Olivera Lab, School of Biological Sciences, University of Utah, Salt Lake City, UT 84112, USA

**Keywords:** alpha-1 antitrypsin, alpha-1 antitrypsin deficiency, descemetocele, LASIK, PRK, SMILE, cornea, corneal erosion, corneal ulcer, refractive surgery

## Abstract

Alpha-1 Antitrypsin Deficiency (AATD) is an autosomal inheritable disorder that impairs the protease inhibitor alpha-1 antitrypsin. This disorder presents with various systemic effects, including liver cirrhosis, centrilobular emphysema, and ocular manifestations. Performing corneal refractive surgery in patients with AATD raises concerns regarding the increased rates of corneal erosions, corneal ulcerations, potential developing descemetoceles, and other ocular manifestations. Patient outcomes for laser-assisted in situ keratomileuses (LASIK), photorefractive keratectomy (PRK), small incision lenticule extraction (SMILE), and other ocular corrective surgeries are lacking in this population. This article provides experiences performing corneal refractive surgery, discusses the current understanding of AATD, including its ocular manifestations, and explores factors to consider when evaluating patients for corneal procedures. The aim of this paper is to address the manifestations of AATD prior to performing corrective vision surgery.

## 1. Introduction

Alpha-1 Antitrypsin Deficiency (AATD) is one of the most common severe systemic conditions worldwide [1]. AATD is a co-dominant autosomal disorder involving the *SERPINA1* gene (previously called *PI* gene) on chromosome 14, which codes for alpha-1 antitrypsin (AAT) [2]. This gene has over 150 different isoforms, many of which cause significant effects on gene expression and function [2,3]. The most common non-disease allele is referred to as “M”, while the most common disease-causing alleles are referred to as “S” and “Z”. These alleles are characterized by their isoelectric points and protease inhibitory capacity. Individuals who are heterozygous (e.g., *Pi*SZ*) or homozygous (e.g., *Pi*ZZ* or *Pi*SS*) for the disease-causing alleles have AATD [2]. AAT is produced in hepatocytes and secreted into the cardiovascular system in unaffected individuals. However, disease-causing alleles lead to misfolded and dysfunctional proteins, which affect the ability of the hepatocytes to secrete AAT, leading to a buildup of misfolded proteins in hepatocytes and eventual chronic hepatocellular apoptosis. Manifestations of AATD include cirrhosis and hepatocellular carcinoma [2,4]. The normal function of AAT involves the breakdown of proteases such as trypsin, neutrophil proteases, and metallomatrix proteases (MMP) [2,5,6]. Thus, a deficiency of AAT leads to the unopposed activation of protease enzymes, which causes a characteristic lung disease and systemic necrotizing vasculitis [2,5,7]. AAT is usually increased during inflammation, decreasing host proteases and infectious proteases. For patients with AATD, these proteases are unregulated and cause more severe inflammatory responses after a triggering event [2,8].

AATD was thought to have been a disease affecting primarily those of Caucasian descent, with approximately 70,000 to 100,000 individuals affected in the United States and Europe, respectively [9,10]. However, recent studies show that AATD exists worldwide, with estimates suggesting that about 3.4 million individuals are affected by this disease [1]. This rate is comparable to cystic fibrosis and suggests that AATD is one of the most common serious hereditary disorders [1]. However, AATD is an underdiagnosed condition, with individuals experiencing very long diagnostic delays [9,10]. On average, individuals first experience manifestations at 39 (±10) years of age and are diagnosed at 45 (±10) years of age [10]. As a result, the manifestations of AATD might be first visualized by an eye care provider.

Myopic patients with AATD may likely request an evaluation for corneal refractive surgeries. Laser-assisted in situ keratomileuses (LASIK), photorefractive keratectomy (PRK), and small incision lenticule extraction (SMILE) are all forms of corneal refractive surgery with established post-operative efficacy, predictability, and safety margins. Studies also frequently compare these surgeries against each other [11]. Despite the extensive literature on AATD, very little literature explores these corneal refractive surgeries in these patients. Furthermore, there are no recommendations regarding corneal refractive surgery or the incidence rate of myopia in this patient population.

The paper aims to discuss the manifestations of AATD, screening prior to corneal refractive surgery, and managing the possible complications. This paper examines our current understanding of the pathophysiology of AATD, the various ocular manifestations of this condition, and important considerations when evaluating affected individuals for potential vision corrective surgery. The data and results of this paper are based on previously conducted studies and do not contain any studies with human participants performed by the authors. However, patients with AATD have been evaluated for LASIK by one of the authors, and the clinical experience in managing these patients is discussed.

## 2. Clinical Experience with LASIK

We evaluated five patients with AATD who were optimal candidates for LASIK surgery. Patients had routine pre-operative workups, and patients were medically cleared by other physicians for LASIK surgery. During refractive surgery screening, the patient is asked about any history of autoimmune diseases, specifically Hashimoto, rheumatoid arthritis, Sjogren, psoriasis, Crohn’s, and ulcerative colitis. After LASIK, all patients were successful with vision correction. Of these, three patients had no further complications. However, two patients had some minor issues after the procedure.

### 2.1. Case Report 1

One patient was a 42-year-old male who presented several days after his LASIK procedure complaining of foreign body sensation, dry eye, fluctuation in vision, and pain in the morning. On slit-lamp examination, the patient was found to have mild corneal abrasion with secondary staphylococcal marginal keratitis and corneal staining at the LASIK flap, bilaterally (Figure 1). The patient was also noted to have “sands of Sahara” diffuse lamellar keratitis (DLK) in the surrounding area. The patient’s uncorrected distance visual acuity (UDVA) and best-corrected distance visual acuity (BDVA) were 20/20 OU. Given the findings, the patient was believed to have concomitant Meibomian gland disease and was treated with one month of doxycycline, topical corticosteroids, and artificial tears. The patient’s symptoms resolved after one month, and UDVA was 20/20. On slit-lamp examination, the patient’s epithelium was well-healed. He was advised to continue with proper eyelid hygiene and warm compress use.

### 2.2. Case Report 2

The other patient was a 38-year-old male who presented one month after LASIK complaining of episodic eye pain, foreign body sensation, and profuse tearing. The UDVA and BDVA were both 20/20 OU. On slit-lamp examination, the patient was noted to have epithelial erosion, dryness, and increased fluoresceine uptake overlying the LASIK flap. Improper remodeling and healing of the corneal epithelium were also noted (Figure 2). The patient was treated for suspected blepharitis with bedtime erythromycin, warm compress, and eyelid hygiene. He had waxing and waning improvement in symptoms that required punctal plugs. The patient was started on doxycycline, and his symptoms improved in 3 months. He was advised to continue to use artificial tears. Both minor complications were successfully treated with proper medical management.

## 3. Ocular Involvement

Maintaining the transparency of the cornea is critical for vision. To keep this transparency, the stromal matrix (composed of collagen and other extracellular proteins) must be kept from the degradation of proteolytic enzymes. Two MMPs known to be critical for remodeling and healing the ocular surface are MMP-2 and MMP-9. These enzymes are upregulated by inflammatory cytokines, suggesting their role in corneal matrix degradation [5]. In fact, patients who present with corneal ulcers are given tetracyclines (e.g., doxycycline) to decrease the activity of MMP [12,13]. Thus, the activity of protease inhibitors is integral to the maintenance of the cornea [14].

AAT is known to exist in all layers of the cornea, with an average level of 29.5 μg in the epithelium, 54.35 μg in the stroma, and 3.55 μg in the endothelium [14]. AAT has also been detected in Descemet’s membrane, aqueous humor, vitreous humor, and tears [14,15,16]. Studies demonstrate that AAT in the cornea is produced independently of the liver [14,16]. Epithelial cells near the tear ducts, stromal cells, and endothelial cells contain AAT mRNA, indicating that AAT is locally produced [14,16]. It is suggested that normal levels of AAT are also responsible for maintaining the integrity of the cornea. For example, a study examining keratoconus found that the AAT concentration was at a level one-sixth of an average human eye [17]. AAT has also been implicated in irreversible corneal graft rejection. Patients who had a reversible graft rejection had a statistically significant higher level of AAT than the irreversible graft rejection group, even when examining total protein between these two groups [16]. The prevalence of AAT in the cornea and its changes in response to stressors highlights its role in maintaining corneal transparency and integrity. Thus, there are notable corneal etiologies when individuals cannot produce AAT.

Ocular manifestations of AATD become evident after infections, surgery, hospitalizations, or other inciting factors. Patients with AATD are more susceptible to developing corneal erosions, a loss of the corneal epithelium. This has been attributed to the decrease in AATD in tear film during times of inflammation [15]. However, as patients with corneal abrasions do not always seek care, cases of AATD-induced recurrent corneal erosions might be misattributed to another etiology. These erosions can progress into ulcerations, as the MMP is uninhibited [18]. Patients with AATD have also been found to have recurrent lower lid styes resulting in multiple, large scars in the inferior area of the cornea in both eyes [19]. This suggests that inflammation of adjacent structures (e.g., eyelid, conjunctiva) allows for proteases to be upregulated in the cornea and cause corneal manifestations in this population. Furthermore, patients with the severe genotype AATD (*Pi*ZZ*) have an earlier onset of these ocular manifestations [19].

Patients with a history of recent hospitalization requiring intubation/sedation have some degree of exposure keratopathy as they do not regularly refresh a protective tear layer. For patients with AATD, the lack of lubrication leads to more severe ocular effects, such as the presence of severe descemetoceles [20]. Treatment for descemetoceles may be limited by the patient’s comorbid medical conditions. Progression of the descemetocele is common and may require therapeutic interventions (e.g., tectonic keratoplasty) [20]. The increased severity of corneal epithelial defects suggests the importance of ophthalmologic evaluation for patients with AATD after a prolonged hospitalization [20]. See Figure 3 for ocular regulation of MMPs.

Uveitis is another ocular manifestation for patients with AATD. A 5-year study examining patients with anterior uveitis reported a high prevalence and incidence of anterior uveitis in patients with a defective allele for AATD (e.g., Pi*MS, Pi*MV, Pi*MZ) [21]. Thus, patients who are heterozygous or homozygous for the *SERPINA1* gene have lower activity of AAT at the site of inflammation. Screening patients for AATD in patients with anterior uveitis of unclear etiology is an ongoing debate as some suggest that AATD may not be the cause [21].

Other ocular manifestations of AATD include the development of vasculitis. Interestingly, patients with granulomatosis with polyangiitis (GPA) showed a significant prevalence of the Pi*Z or Pi*S allele, suggesting overexpression of either allele is associated with AATD. Given this close association, approximately 7% of patients with GPA have AATD, where GPA is considered an AATD manifestation, similar to cirrhosis and emphysema [22]. Patients with GPA have manifestations that include conjunctivitis, episcleritis, keratitis, retinal vasculitis, and vascular obstruction [23]. Thus, patients with AATD should be evaluated for similar presentations to GPA.

An increased rate of optic neuropathy was also found in patients with AATD. Younger et al. found that patients with optic neuropathy had decreased levels of AAT. AAT might have a role in preventing ischemia of the optic nerve where low serum rates may lead to optic neuropathy. Thus, patients with AATD should be evaluated for clinical manifestations of optic neuropathy [24].

## 4. Considerations

When evaluating patients with AATD for corrective vision surgeries, specific considerations should be taken. Although a medical history of AATD is not a contraindication for surgery, the buildup of misfolded protein in the liver progresses to cirrhosis, requiring further risk stratification. A hepatologist should routinely evaluate patients with this history to ensure that liver function is stable [25]. Patients who are starting to show symptoms of AATD or have a history of AATD in the third or fourth decade of life should be evaluated for potential corneal manifestations. Table 1 and Table 2 summarize suggested questions and tests to evaluate patients with AATD for corneal refraction procedures.

The only definitive treatment for AATD is a liver transplant, which corrects the deficient AAT in the vascular supply. Patients with AATD represent 1% of the adult liver transplant list and 3.5% of the pediatric liver transplant list [26]. Given the likelihood of receiving transplantation, discussion of the mortality rate after transplantation is essential. The 5-year survival rate is 83% in adults and 90% in children [26]. In addition to liver transplants, these patients also receive lung transplants depending on the severity of their disease [26].

For patients with AATD who receive transplants, it is vital to consider the ocular manifestations of chronic immunosuppression. Patients who have received transplants are at significantly increased risk of opportunistic infections. Reported infections include herpetic viral retinitis, Cytomegalovirus retinitis, and fungal chorioretinitis [27]. Other manifestations of chronic immunosuppression are herpetic keratitis, choroidal pseudolymphoma, central retinal vein occlusion, dacryocystitis, and cyclosporin toxicity [27]. Patients with severe pulmonary manifestations of AATD and an inability to lie flat should undergo a pulmonary evaluation [28].

Patients with AATD should also be counseled on the surgical outcomes of corrective vision surgeries. It is essential to consider the incidence of dry eye syndrome after corrective vision surgery. LASIK is known to cause mechanical injury to the corneal nerves, which affects reflexive tear secretion [29]. As patients already have a risk of developing dry eye after LASIK, patients with AATD may be at an increased risk of developing corneal erosion and ulcerations due to the lack of AAT. See Table 1 for important questions to ask to assess LASIK candidacy.

## 5. Conclusions

The ocular manifestations of AATD include ulceration, erosions, descemetoceles, anterior uveitis, and retinal vasculitis. To the authors’ knowledge, there are no published case reports of LASIK, PRK, or SMILE in patients known to have AATD. The authors report successful LASIK procedures on individuals with AATD. Patients with self-limiting post-operative complications were successfully managed. However, a lack of defined guidelines for AATD makes it challenging to determine if LASIK, PRK, or SMILE can be safely performed in this patient population. Patients should be counseled on the risks and benefits of refractive surgery, including suboptimal surgical outcomes, especially given their history [30].

## Figures and Tables

**Figure 1 jcm-11-04175-f001:**
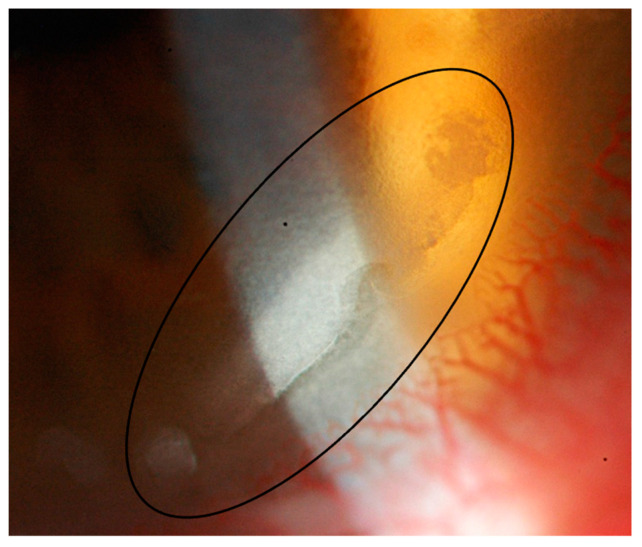
Staphylococcal marginal keratitis in the area of the LASIK flap (black circle) with surrounding DLK.

**Figure 2 jcm-11-04175-f002:**
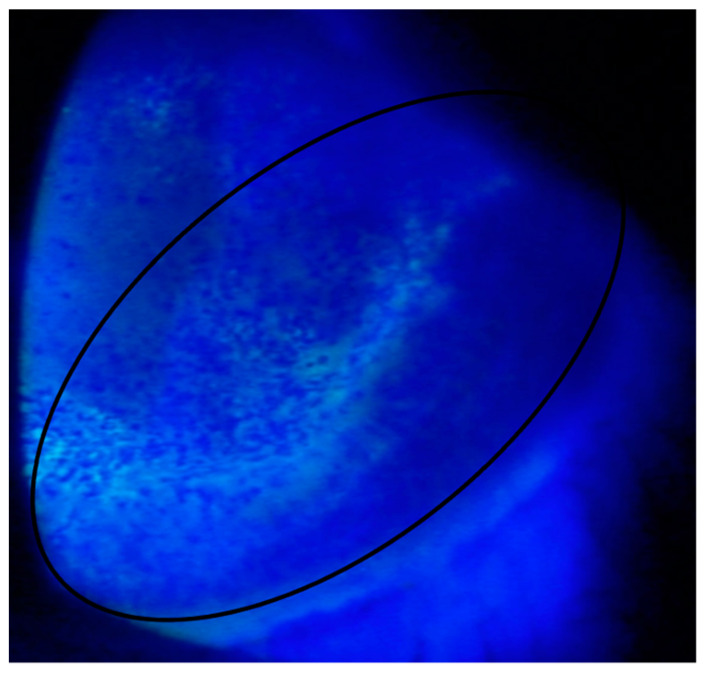
Corneal erosion with poor epithelial remodeling/healing overlying the LASIK flap (black circle).

**Figure 3 jcm-11-04175-f003:**
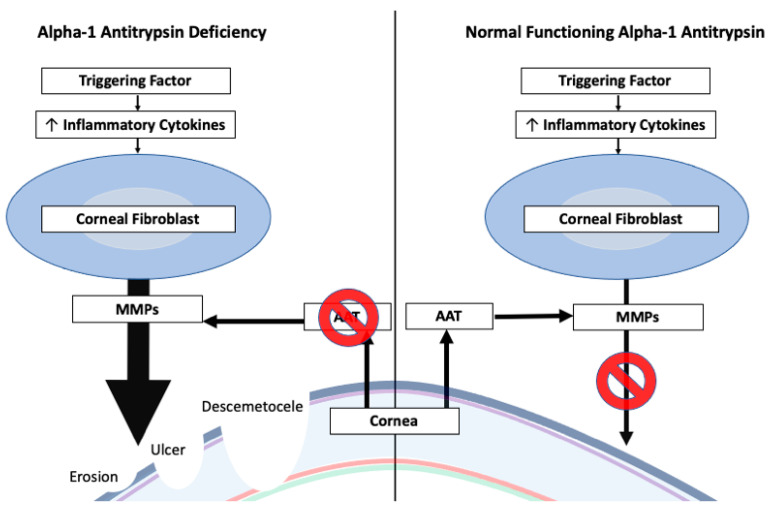
Dysregulation of Corneal MMPs Leading to Corneal Pathology. The increasing thickness of the arrow represents a greater amount of MMPs. Legend: Metallomatrix Protease (MMP), Alpha-1 Antitrypsin (AAT).

**Table 1 jcm-11-04175-t001:** Suggested questions in an ophthalmic evaluation for patients with Alpha-1 Antitrypsin Deficiency (AATD).

**Patient History**	Do you feel like your eyes are dry or have a foreign body sensation?
	Do you have a history of dry eyes or recurrent corneal abrasions?
	Do you have any history of ocular infection?
	What is your smoking history?
	Any vaping or second-hand smoke exposure?
	Any occupational hazards that can cause eye irritation? When were you diagnosed with AATD?
	Why were you evaluated for AATD?
**Review of Systems**	Based on previous symptoms or family history?
	Are you regularly evaluated by a hepatologist, or another physician, to manage your liver?
	Have your liver function tests (LFTs) been stable?
	Have you been informed about signs of cirrhosis by any provider?
	Do you have any difficulty breathing at rest?
	Has a pulmonologist evaluated you for respiratory issues?
	If so, what is the state of your lungs?
	Any other AATD-related conditions (e.g., vasculitis, nephropathy, inflammatory bowel disease, psoriasis) and how are they managed?
	Are you on a liver or lung transplant list?
	Any history of ICU admissions or prior intubation?
**Medications**	Are you on any medications for AATD or other comorbid conditions (e.g., prednisone, augmentation therapy, antibiotics, chronic obstructive pulmonary disease (COPD) medication)?

**Table 2 jcm-11-04175-t002:** Suggested clinical testing for patients with Alpha 1 Anti-trypsin Deficiency (AATD).

Testing:	
Ocular Surface Integrity	Schirmer Testing, Fluorescein dye, tear film analysis, tear osmolarity, signs of early corneal erosions
Visual Assessment	Comprehensive dilated ophthalmic examination
	Indirect ophthalmic examination for retinal vasculitis
	Visual acuity
	Evaluation of superficial punctate keratitis
	Evaluation of keratic precipitates
	Evaluation of meibomian gland disease
	Inflammatory dry eye
Imaging	Optical Coherence Tomography (OCT) of the optic nerve to assess for neuropathy
	Placido Corneal topography to evaluate integrity of anterior corneal surface and uniformity of the mires
	Fluorescein Angiogram and OCT to assess for peripheral retinal vasculitis

## Data Availability

Data sharing does not apply to this article as no datasets were generated or analyzed during the current study.

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
