# Peer review of "Assessing Patients with Alpha-1 Antitrypsin Deficiency for Corneal Refractive Surgery: A Review and Clinical Experience"

_jcm, 2022, doi:10.3390/jcm11144175_

Round 1

Reviewer 1 Report

The authors analyze patients with alpha 1 antitrypsin deficiency for corneal refractive surgery and they found that performing corneal refractive surgery, discuses the current understanding of AATD, including its ocular manifestations, and explores factors to consider when evaluating patients for corneal procedures.

The topic of the manuscript is very original and could be of interest of the potential ophthalmologist and optometrist scientific reader community. Prior to continue with the publication process, son issues must be raised and explain.

The title should resume the findings and catch the audience with some scientific soundness words.

In the abstract and at the end of the introduction the objective or aim is missing and authors should include it in both places.

Place the citation to the references always prior to the punctuation sign.

Explain the AATD in a paragraph, the difference corneal refractive surgery in other paragraph, the link between both ideas in other, the lack of scientific content following it and at the end, finish the introduction with the objective or aim of the research.

Include some references about the efficacy of corneal laser very common, such SMILE.

This is the first time I review a perspective article, I supposed that the journal approve the kind of submission, since is not a review or original, I do not know how really it you fit on the journal of clinical medicine overview.

In the second section case report were explain, so if could be possible state this on the title of the section and subheading the case report to recognize this part.

How really do you know that the clinical finding of the sand of Sahara match with the AATD issues?

Do you think that the eyelid hygiene could alter the DLK after the good healing?

Improve the  quality and the letter size of the figure 3, it is very difficult to see in a proper way.

In the suggest questions or testing, summarize the table and only report the problem to ask about or the test to do but with less and simple word. It is not necessary to write down the complete question in order a reader only have to read it.

If it is possible, separate the table 1 into 2 tables one for the questions and the other for the test, because the visual appearance is very poor.

At the end, on the conclusion, you can not give general conclusions for a perspective manuscript with only two case report. If you want to give authors opinion or the limited experience but I do not recommend to give general advise or to establish some conclusions due the lack of findings in this research.

Reviewer 2 Report

The authors reported the patient outcomes for laser-assisted in situ keratomileuses (LASIK), photorefractive keratectomy (PRK), small incision lenticule extraction (SMILE),This article provides experiences performing corneal refractive surgery,  and explores factors to consider when evaluating patients for corneal procedures.

Both of the first two patients develpled peripherial cornea inflammation, especially the first one develpoed ulceration and DLK, thus did the authors examined the patient for autoimmune disease , which could also induce periferial keratitis often.

The  38-year-old male  patient was started on doxycycline, and his symptoms improved in 3 months. He was advised to continue to use artificial tears. Did the authors test the patients for dry eye disease, especially for Sjogren's Sndrome that also easy to develop a epithelial defect?

Round 2

Reviewer 1 Report

Comments solved